# Immune-Related Thyroiditis as a Predictor for Survival in Metastatic Renal Cell Carcinoma

**DOI:** 10.3390/cancers14040875

**Published:** 2022-02-10

**Authors:** Shira Sagie, Moran Gadot, Meital Levartovsky, Hadas Gantz Sorotsky, Raanan Berger, Michal Sarfaty, Ruth Percik

**Affiliations:** 1Institute of Oncology, Sheba Medical Center, Ramat Gan 52621, Israel; shirasagie@gmail.com (S.S.); moran.gadot@sheba.health.gov.il (M.G.); meital.levartovsky@sheba.health.gov.il (M.L.); hadas.gantzsorotsky@sheba.health.gov.il (H.G.S.); raanan.berger@sheba.health.gov.il (R.B.); michalchen1@gmail.com (M.S.); 2Sackler Faculty of Medicine, Tel Aviv University, Tel Aviv 6997801, Israel; 3The Sheba Talpiot Medical Leadership Program, Sheba Medical Center, Ramat Gan 52621, Israel; 4Division of Endocrinology, Diabetes and Metabolism, Sheba Medical Center, Ramat Gan 52621, Israel

**Keywords:** immune-mediated adverse events, immunotherapy, renal cell carcinoma, thyroiditis

## Abstract

**Simple Summary:**

In this study, we evaluated the association of immune mediated thyroid dysfunction (irT) with survival in 123 metastatic renal cell carcinoma patients treated with immunotherapy in a single center. We found that irT is a prevalent and early event associated with prolonged survival in high-risk patients.

**Abstract:**

Immune checkpoint inhibitors (CPI) are indicated for metastatic renal cell carcinoma (mRCC). Immune-related thyroiditis (irT), an immune-related adverse event (irAE), affects up to 30% of patients. We aimed to determine whether irT is associated with overall survival in mRCC. A retrospective cohort study of 123 consecutive patients treated with CPI for mRCC in a single center between 2015 and 2020 was conducted. Disease risk stratification was assessed by two methods: Heng criteria and a novel dichotomic stratification system to “Low risk” versus “High risk” adding number of metastatic sites. Thirty-eight percent of patients developed irT. In the general cohort, irT was not associated with a survival benefit. However, irT was associated with better survival in the poor risk group per Heng criteria (*n* = 17, HR = 0.25, *p* = 0.04) and in the novel “High risk” group (HR = 0.28, *n* = 42, *p* = 0.01), including after accounting for covariates in multivariate analysis (HR = 0.27, *p* = 0.003). Having any irAE was associated with improved survival in the whole cohort, with no significant correlation of any specific irAE, in either the whole cohort or the “High risk” group. We conclude that irT is an early and prevalent irAE, associated with prolonged survival in patients with poor/“High” risk mRCC.

## 1. Introduction

The extensive integration of immunotherapy has revolutionized cancer treatment, while introducing a heterogeneous class of immune-related adverse events (irAEs) [1]. Checkpoint inhibitors (CPI) have routinely been used in the treatment of metastatic renal cell carcinoma (mRCC) since 2015, first as a monotherapy in subsequent lines and currently in the first-line setting as part of combination therapy with either a vascular endothelial growth factor inhibitor (VEGFi) or a second CPI (nivolumab with ipilimumab) [2]. The risk stratification model for mRCC patients in the VEGFi era, presented by Heng et al., stratified patients according to overall survival (OS) [3,4]. The model divided patients into favorable-, intermediate- and poor-risk groups according to six prognostic factors. This clinical model has been used for risk stratification in all recent immunotherapy phase III trials [2] and predicted the superiority of immunotherapy combinations versus sunitinib in intermediate–poor-risk patients. Other biomarkers to predict responses to CPI are still under research.

Interim treatment outcomes such as major radiological responses or high grade irAEs, that may indicate activation of the patient’s immune system, have been associated with a longer duration of response and improved overall survival (OS) [5,6,7].

Immune-related thyroiditis (irT) is a prevalent irAE, affecting 6–30% of CPI-treated patients, with an increased risk among patients with renal cell carcinoma [1,8,9,10,11]. This may be related to pre-existing thyroid injury induced by tyrosine kinase inhibitors [12]. Cytotoxic T-cells infiltrating the thyroid gland, inducing inflammation and necrosis play the dominant role in the pathogenesis of irT [13,14,15]. However, a humoral component resembling classic Hashimoto’s disease is also assumed to exist, based on a higher prevalence of irT in patients with pre-existing anti-thyroid peroxidase (TPO) antibodies [16,17].

In a previous study, we found an association between irT and survival in a small cohort of RCC patients (*n* = 27) but not in melanoma, lung cancer or bladder cancer patients [12]. The goal of the present study was to define the association between irT and overall survival in patients with renal cell carcinoma treated with immunotherapy.

## 2. Materials and Methods

### 2.1. Study Population and Design

This retrospective cohort study is based on a database of consecutive patients treated with immunotherapy at Sheba Medical Center between January 2015 and December 2020. Patients were included if they were treated for mRCC with CPI as monotherapy or combination therapy, with either nivolumab, pembrolizumab, avelumab or ipilimumab, for any line of treatment as standard of care or as part of a clinical trial (*n* = 127). Data were retrieved from the patients’ electronic records, including demographics, tumor and treatment details, thyroid hormone measures and replacement therapy status. The study was approved by the Sheba Medical Center institutional ethical board. Requirement for consent was waived by the ethics committee as the research was deidentified.

### 2.2. Study Variables

The study population was categorized by immune-related thyroiditis (irT) status. Thyroid function tests, including serum T4 and TSH, were analyzed at three time frames: baseline (i.e., at RCC diagnosis), after treatment with previous line VEGFi and after CPI treatment. irT was defined as when either fT4 or TSH was unequivocally outside the normal range (serum TSH above 8 mIU/L or below 0.4 mIU/L, serum fT4 above 18 pmol/L or below 7 pmol/L), or when levothyroxine replacement therapy was initiated. irT was defined only if detected during the period of immunotherapy or in close proximity to it.

Overall survival (OS) was calculated from the first CPI treatment. Data cut-off point was 21 May 2021.

Additional data collected included patients’ demographics (sex, age, smoking status, and background autoimmune diseases); disease characteristics (malignancy diagnosis date, primary metastatic disease diagnosis date, nephrectomy status, histology, presence of sarcomatoid features, metastatic sites at diagnosis, and scoring for Heng criteria (neutrophilia, thrombophilia, anemia, less than 1 year between diagnosis and treatment beginning, hypercalcemia, and ECOG performance status)). Additional data included: first-line treatment date, previous treatment lines, additional immune-related adverse events (irAE), use of high dose steroids due to an irAE, time from CPI treatment onset to thyroid dysfunction, treatment type, and best response to CPI treatment.

### 2.3. Statistical Analysis

The study sample was described using means and standard deviation (SD) for continuous variables, medians and interquartile range (IQR) for variables with skewed distribution, and frequencies and percentages for categorical variables. The unadjusted associations of demographics, disease characteristics and other independent variables with thyroid dysfunction status were examined using Student’s *t*-test for continuous variables that followed a normal distribution, and the Mann–Whitney *U* test for variables that did not follow a normal distribution, and the chi-squared test and Fisher’s exact test were used as appropriate for categorical variables.

Kaplan–Meier curves were used to compare OS between the two study groups. Cox proportional hazard regression models were used to estimate hazard ratios (HRs) with 95% confidence intervals (CIs) for OS for both univariate and multivariate models. Interactions of suspected disease prognostic factors with irT on OS were studied in separate Cox proportional hazard regression models. *p* < 0.05 was considered statistically significant. Missing data: we were able to obtain 93% of all datapoints. Only four patients were excluded because they lacked irT status, as all patients must have thyroid functional tests recorded in the beginning of each treatment cycle in our institution. Two patients lacked data needed for classification to risk groups. The statistical analyses were performed using R version 3.4.2.

## 3. Results

### 3.1. Baseline Characteristics

A total of 127 patients were treated with immunotherapy; of them, 123 had data on irT development and were included in the study, 70.7% were male, the median age was 62, and 86.6% had clear cell histology. Nearly half of the patients were treated with an ipilimumab–nivolumab combination, 28.5% were treated with single agent nivolumab and the remainder with VEGFi-CPI combinations; 52% had received VEGFi in previous treatment lines.

Forty-seven patients (38%) developed irT. Patients’ characteristics according to their irT status are presented in Table 1. There were no significant differences in age, sex, pathological subtype, risk groups, or number of metastatic sites between the groups. In addition, rates of previous thyroid dysfunction at baseline or after previous treatment with VEGFi did not statistically differ between the groups. The kind of CPI, receiving a dual CPI combination, or receiving a combination of a CPI and a VEGF inhibitor—all were not associated with the irT outcome (Table 1 and additional post hoc tests).

### 3.2. Thyroid Hormone Dynamics

TSH suppression occurred at a median of 42 days, T4 elevation at a median of 66 days and TSH exceeding 8 mIU/L at a median of 124 days. All patients with irT had documented elevated TSH before thyroid replacement treatment was initiated (median 14.35 mIU/L). A thyrotoxic phase (median T4 level 20 pmol/L) was documented in 26 patients (55% of irT patients).

### 3.3. Immune Checkpoint Induced Thyroiditis as a Predictor for Survival in RCC Patients

With a median follow up of 17 months from treatment initiation, 65 (53%) patients were still alive at data collection. While no association was found between irT and overall survival in the entire cohort (Figure 1A), there was a significant interaction when dividing into Heng’s risk groups (*p* = 0.0251, Figure 1B–D). irT positively correlated with survival in the poor risk group (*n* = 17, HR = 0.25 *p* = 0.04) but negatively correlated with survival in the favorable risk group (*n* = 24, HR = 3.8 *p* = 0.04). No significant correlation was found in the intermediate risk group (*n* = 81, HR = 0.63 *p* = 0.23).

As the sample size of each risk group was small, we evaluated these findings in a more robust sample size by dividing our cohort into two larger risk groups, instead of three: a new high-risk group, which included the poor risk patients by Heng criteria plus the intermediate risk patients with more than two metastatic sites (*n* = 42), and a new low-risk group, which included the Heng favorable group plus the intermediate risk patients with up to two metastatic sites (*n* = 80).

In the new high-risk group, 40.5% had thyroiditis, and irT was associated with better survival (HR = 0.28, *p* = 0.01, Figure 2A), while no such association was found in the low-risk group (*p* = 0.38, Figure 2B).

In order to verify the new classification variable, OS since CPI initiation was analyzed by Cox proportional hazards regression models; the new high-risk group had an HR of 2.85 (CI 1.7–4.8, *p* < 0.001) compared to the new low-risk group (Appendix A). The new classification variable also had a strong interaction with irT (*p* = 0.007).

### 3.4. Prevalence of Other Immune-Related Adverse Events and Effect on Survival

Forty-seven percent of all patients had at least one irAE other than irT.

Having an irAE other than irT was significantly associated with improved survival in the entire cohort (HR = 0.34, *p* < 0.001) but not in the risk subgroups. Treatment with high dose steroids (defined as >40 mg/day of prednisone or equivalent) was prevalent in 29% of our cohort and was also associated with improved survival in the entire cohort but not in the high-risk group. irT was the most prevalent irAE (38%). No statistically significant difference was found in any specific irAE between the patients receiving Ipi–Nivo to the patients receiving other treatments, but a trend of higher prevalence in the Ipi–Nivo group was noted for pruritus, arthritis, hepatitis, pneumonitis, colitis and hypoadrenalism (Appendix A).

No other irAE had a significant effect on survival in the entire cohort or in the high- risk group (Table 2).

In multivariate analyses irT continued to be the strongest significant protective factor for survival in the high-risk group (considering: age, sex, other irAE, history of previous thyroid dysfunction, previous treatment lines, and treatment type) with a multivariate adjusted HR of 0.27 (*p* = 0.003).

## 4. Discussion

Our study focused on the association of irT with prognosis in mRCC patients, a population with a high frequency of this irAE. To our knowledge, this is the largest cohort studied to evaluate irT in mRCC patients to date. Several studies have evaluated the prognostic value of irT in several cancer types with indeterminate findings [7,8,10,12,18,19,20]. We found a strong and significant association with survival in poor risk/High risk patients.

### 4.1. irT among Patients with mRCC

In our cohort, 38% of patients developed thyroid dysfunction. Previously, it was suggested that treatment with VEGFi is an inducer of thyroid injury, increasing the risk for irT with CPI exposure [12]. While the rate of irT is higher in our cohort than the commonly reported 20–30% [7,8,10,12,18,19,20], the incidence rate did not differ between patients who were treated with immunotherapy in the first-line treatment or after previous treatment with VEGFi. For patients with previous thyroid dysfunction on VEGFi we found a non-significant trend of increased irT risk (HR = 2.14, *p* = 0.08).

By studying the thyroid hormone levels in three different time frames (at baseline, following VEGFi, and after CPI) we were able to optimize our definition of irT and understand its risk factors. We found no statistically significant effect of previous thyroid disease, autoimmune disease or previous treatment line. Moreover, irT was a prevalent phenomenon in all RCC subtypes, all risk groups and regardless of which treatment combination was used (Table 1). A past comparison of patients with RCC and gastrointestinal stromal tumors treated with sunitinib also demonstrated a higher rate of thyroid dysfunction in the RCC patients [21]. A possible explanation for the increased irT among patients with RCC is its immunogenic properties and the high rate of paraneoplastic syndromes, although not specifically of the thyroid gland [22]. The relationship between these phenomena and treatment-related thyroid dysfunction in RCC patients requires further research. The timing of thyroid dysfunction was similar to that reported in previous studies [12,19].

### 4.2. irT and Survival

In our cohort (*n* = 123) we did not find an association of irT with OS. Prior studies have shown mixed results; two smaller studies found a protective association of irT on OS [7,12]. In contrast, another study found no such association in RCC with the limitation that other urologic malignancies were grouped together, despite the significant variability in clinical, molecular, and pathological characteristics [18]. Several studies in melanoma demonstrated that irT is predictive for survival [19], while others did not find any association [11].

Our relatively large cohort with a single malignancy type allowed a thorough investigation of the association of irT with survival in risk subgroups, an analysis not performed in previous studies. We found that theHeng risk criteria interacts in the association of irT with survival. In the poor-risk group, there was a strong predictive effect of irT on survival (*n* = 17, HR = 0.25, *p* = 0.04) whereas an opposite effect was observed in the favorable-risk group (*n* = 24, HR = 3.8, *p* = 0.04). Most patients were included in the intermediate-risk group, where no significant effect was observed. Thus, we separated the cohort into two risk groups.

When we analyzed the survival differences between the three Heng risk groups, there was no significant difference in survival between the favorable and intermediate groups (HR = 1.15, *p* = 0.697). Separation into two risk groups of intermediate–poor and favorable, as performed in recent clinical trials, resulted in a non-significant deviation of the survival curves (*p* = 0.24) (results not shown). Hence, we evaluated additional factors that could separate the groups into two risk groups. The number of metastatic sites is a known prognostic factor in mRCC and was included in previous scoring systems, as in the French classification and the IKCWG classification [3,23,24]; it was chosen for splitting the intermediate group as it was both a strong prognostic factor in our cohort (HR = 2.2 *p* < 0.001) in uni- and multi-variate analysis and had a strong interaction with irT (*p* = 0.0085). A similar classification referred to as “favorable/non-favorable” was used in the prospective active surveillance trial due to the lack of prognostic value of the known risk groups in their cohort [25].

A possible explanation for the interaction of our new high/low-risk classification with irT effect on survival may be related to tumor burden. Several studies indicated that tumor burden and number of metastatic sites are negative predictive factors for immunotherapy efficacy [26,27,28]. It is suggested that the immune penetrance is lower when the tumor burden is high and that advanced tumors’ microenvironments have greater populations of immune suppressive cells [26,27]. Thus, in the high-risk group with a high tumor burden and lower chance of immune activation, the subgroup that was able to effectively activate their immune system (as seen by the surrogate marker of irT) had improved survival prospects.

### 4.3. Other irAE

Following the observed protective effect of irT on survival in the high-risk group patients, we studied the prevalence and effect on survival of additional immune-related adverse events (irAE) in the whole population and in the high-risk group. Having any non-thyroid irAE had a protective effect in the whole cohort as in previous reports [6,7,10], as did a history of treatment with high-dose steroids for irAE; this is in agreement with a recent report [29]. Other specific irAEs were not associated with survival in our cohort, possibly due to their low frequency.

### 4.4. Study Limitations

This was a retrospective study. The median follow-up was 17 months after treatment initiation; a longer follow-up may be needed to fully evaluate the effect of irT on survival in the low-risk group as most (58%) of the patients in this group were still alive at the data cutoff point. While our group was homogenous with regard to the primary tumor, the treatment line sequence was variable. This was adjusted in multivariate analyses, but remaining confounding factors could exist. Non-thyroid-related low-grade irAEs may have been under-reported if they did not have clinical significance as per the retrospective nature of this study. The use of high-dose steroids (which is well documented) allowed us to identify all clinically significant high grade irAEs. Lastly, we did not have levels of anti TPO antibodies, which could have shed more light on the irT mechanism, and differences between risk groups.

## 5. Conclusions

Our study evaluated irT, a prevalent and early irAE in patients with mRCC, suggesting that irT is a predictive marker in high-risk patients but not in low-risk patients, a subject for further research.

## Figures and Tables

**Figure 1 cancers-14-00875-f001:**
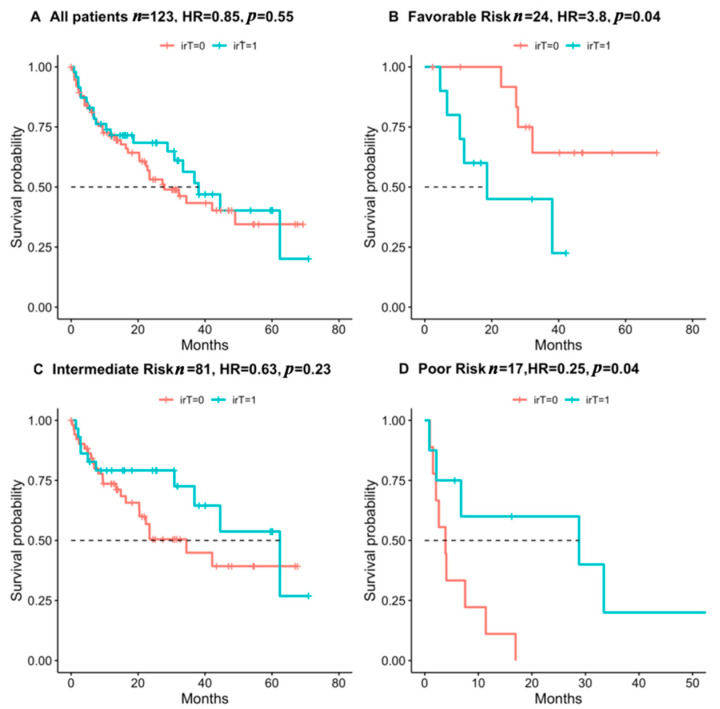
Association of irT with survival in all patients and by Heng risk groups. Kaplan–Meier plots of overall survival. HR and *p*-values were calculated by Cox proportional hazard regression models.

**Figure 2 cancers-14-00875-f002:**
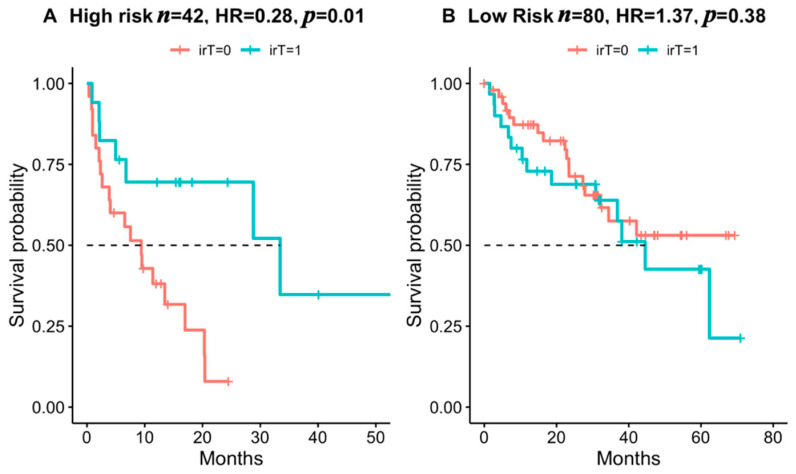
Association of irT with survival probability by new risk groups. Kaplan–Meier plots of overall survival. HR and *p*-values calculated by Cox proportional hazard regression models.

**Table 1 cancers-14-00875-t001:** Patient characteristics of study population divided by irT status.

Variables	Overall	Immune-Related Thyroiditis	No Immune-Related Thyroiditis	*p*
*N* (%)	123 (100)	47 (38)	76 (62)	
Gender, male (%)	87 (70.7)	31 (66)	56 (73.7)	0.477
Age at diagnosis (median (IQR))	62 (53, 69)	65 (58, 69)	61 (51, 69)	0.110
Background autoimmunity (%)	8 (6.5)	2 (4.3)	6 (7.9)	0.675
Background thyroid dysfunction (%)	11 (8.9)	6 (12.8)	5 (6.6)	0.399
RCC subtype (%)				0.726
Clear Cell	103 (86.6)	41 (87.2)	62 (86.1)	
Other	20 (13.3)	6 (12.8)	14 (13.9)	
Number of metastatic sites (median (IQR))	2 (1, 3)	2 (1, 3)	2 (1, 3)	0.814
Heng risk group (%)				0.645
Favorable	24 (19.7)	10 (21.3)	14 (18.7)	
Intermediate	81 (66.4)	29 (61.7)	52 (69.3)	
Poor	17 (13.9)	8 (17.0)	9 (12.0)	
Received prior VEGFi (%)	64 (52)	25 (53.2)	39 (51.3)	0.987
Thyroid dysfunction after VEGFi treatment (%)	29 (23.8)	15 (32.6)	14 (18.4)	0.118
CPI type (%)				0.047
Ipilimumab–Nivolumab	60 (48.8)	22 (46.8)	38 (50.0)	
Nivolumab	35 (28.5)	12 (25.5)	23 (30.3)	
Pembrolizumab-Axitinib	17 (13.8)	7 (14.9)	10 (13.2)	
Avelumab-Axitinib	6 (4.9)	1 (2.1)	5 (6.6)	
Other	5 (4.1)	5 (10.6)	0 (0.0)	

RCC = Renal Cell Carcinoma, VEGFi = Vascular Endothelial Growth Factor inhibitor, CPI = Checkpoint Inhibitor, IQR = Interquartile range. Four patients were not classified by irT due to missing data.

**Table 2 cancers-14-00875-t002:** Immune-related adverse events prevalence and effect on survival in the study population vand in the high-risk group.

Variables	All Patients, *n* (%)	Univariable HR (CI, *p* Value)	High Risk Patients, *n* (%)	Univariable HR (CI, *p* Value)
*N*	123		42	
Immune-related adverse events other than irT, any	57 (47.1)	0.34 (0.20–0.59, *p* < 0.001)	15 (36.6)	0.47 (0.19–1.15, *p* = 0.098)
Treated by high dose steroids	37 (30.1)	0.48 (0.26–0.89, *p* = 0.020)	9 (21.4)	0.74 (0.25–2.18, *p* = 0.589)
Immune-related thyroiditis	47 (38.2)	0.85 (0.50–1.45, *p* = 0.550)	17 (40.5)	0.28 (0.10–0.76, *p* = 0.012)
Encephalitis	2 (1.6)	NA	2 (4.8)	NA
Pruritus	9 (7.3)	0.42 (0.13–1.36, *p* = 0.149)	2 (4.8)	NA
Rash	9 (7.3)	0.28 (0.07–1.14, *p* = 0.076)	1 (2.4)	NA
Neuropathy	3 (2.4)	NA	1 (2.4)	NA
Nephritis	6 (4.9)	0.44 (0.11–1.80, *p* = 0.253)	1 (2.4)	NA
Arthritis or myositis	4 (3.3)	NA	1 (2.4)	NA
Hepatitis	15 (12.2)	0.62 (0.25–1.55, *p* = 0.304)	5 (11.9)	0.55 (0.13–2.34, *p* = 0.417)
Pneumonitis	7 (5.7)	0.47 (0.11–1.91, *p* = 0.289)	1 (2.4)	NA
Diarrhea or colitis or gastritis	15 (12.2)	0.46 (0.18–1.15, *p* = 0.096)	2 (4.8)	NA
Hypoadrenalism	3 (2.4)	NA	3 (7.1)	NA

Univariate Hazard Ratios (HR) were calculated separately for each irAE. High dose steroids were defined as above 40 mg prednisone or equivalent a day.

## Data Availability

Deidentified data will be offered on request.

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
