# Peer review of "Immune-Related Thyroiditis as a Predictor for Survival in Metastatic Renal Cell Carcinoma"

_cancers, 2022, doi:10.3390/cancers14040875_

Round 1
Reviewer 1 Report
The Authors, in their work, analysed if immune-related thyroiditis developed in mRCC patients after treatment with immune checkpoint inhibitors correlated with the overall survival of these patients. The topic of the manuscript is attractive; however, some points need clarification:
- The authors should discuss in more detail whether there is an association between the type of used CPI in mRCC treatment and the development of irT. Does any CPI contribute more to irT development?
- As the authors wrote themselves, a limitation of the work is the lack of information on anti-TPO and anti-TG antibodies levels.
- The levels of T4 and TSH of patients should be given in table 1.
- Some grammar and spelling mistakes.
Reviewer 2 Report
This is a report by Dr. Shira Sagie et al. on the association of thyroiditis as an immune-related adverse event with response to immune checkpoint inhibitors in renal cancer.
It is well known that immune-related adverse events correlate with immune checkpoint inhibitor response. In the present study, they stratified patients according to the risk classification of renal cancer and found that there was positive correlation with thyroiditis in the intermediate risk and poor risk groups, and negative correlation in the favorable risk group, albeit there was no association in the overall population. It would be interesting to see if T-cell immunity induced by immune checkpoint inhibitors produces different results in terms of antitumor effects in different risk categories. However, this study did not examine the mechanism of T-cell immunity, and there are concerns about the survival analysis.
- The data cutoff point for this study is not provided. The OS is based on CPI treatment, and the mOS of the high risk group in Supplement 1 is nearly 100 months, even though the patients were treated consecutively with CPI from 2015-2020.
- The OS of the high-risk group in Fig. 2 is inferior to that of the low-risk group (mOS of irT=1 is about 30 month in high risk group, however, mOS of irT=1 is over 40 month in low risk group), whereas the OS of the high-risk group in Supplement 1 is significantly superior and mOS is nearly 100 month. What is the reason for this difference?
- About half of the patients were treated with Ipi+Nivo. It has been shown in many clinical trials that concomitant use of anti-CTLA-4 antibody alters the irAE profile. The present study should show how the effect was associated with pituitary hypoadrenocorticism and skin disorders, which are characteristic of Ipi+Nivo and other treatments separately or Ipi+Nivo.
- In Table 2, any irAE excludes thyroiditis? 38.2% of the patients had iT, and it is thought that there are overlapping cases of any irAE with iT, but did the overlapping cases have a good prognosis in any irAE+ even in the favorable group?
Round 2
Reviewer 2 Report
My concerns have been adequately addressed and mistakes have been corrected.